



**Observations of iodine monoxide over three summers at the Indian Antarctic bases, Bharati**
**and Maitri**
Anoop S. Mahajan[1*], Mriganka S. Biswas[1,2], Steffen Beirle[3], Thomas Wagner[3], Anja Schönhardt[4],
Nuria Benavent[5] and Alfonso Saiz-Lopez[5]
[1]Centre for Climate Change Research, Indian Institute of Tropical Meteorology, Ministry of Earth
Sciences, Pune, 411008 India.
[2]Savitribai Phule Pune University, Pune, 411008 India.
[3]Max-Planck-Institut für Chemie (MPI-C), Satellitenfernerkundung, 55128 Mainz, Germany
[4]Institute of Environmental Physics, Department of Physics and Electrical Engineering, University
of Bremen, Bremen, 330440 Germany.
[5]Department of Atmospheric Chemistry and Climate, Institute of Physical Chemistry Rocasolano,
CSIC, Madrid 28006, Spain
* corresponding author: Anoop S. Mahajan (anoop@tropmet.res.in)





**Abstract**
Iodine plays a vital role in oxidation chemistry over Antarctica, with past observations showing
highly elevated levels of iodine oxide (IO) leading to severe depletion of boundary layer ozone in
West Antarctica. Here, we present multi axis differential absorption spectroscopy (MAX-DOAS)
based observations of IO over three summers (2015-2017) at the Indian Antarctic bases, Bharati
and Maitri. IO was observed during all the campaigns, with mixing ratios below 2 pptv for the
three summers, which are lower than the peak levels observed in West Antarctica. This suggests
that sources in West Antarctica are different or stronger than sources of iodine compounds in East
Antarctica. Vertical profiles estimated using a profile retrieval algorithm showed decreasing
gradients, with a peak in the lower boundary layer. The ground-based instrument retrieved vertical
column densities (VCDs) were approximately a factor of three-five higher than the VCDs reported
using satellite-based instruments, which is most likely related to the sensitivies of the
measurement techniques. Airmass back-trajectory analysis failed to highlight a source region, with
most of the airmasses coming from coastal or continental regions. This study highlights the
variation in iodine chemistry in different regions in Antarctica and the importance of a long-term
dataset to validate models estimating the impacts of iodine chemistry.
Keywords: Iodine; Antarctica; halogens; DOAS





## 1. Introduction

Observations of reactive halogen species (RHS) have been made in the Antarctic marine boundary layer (MBL) for almost two decades. Early observations focused on bromine oxide (BrO), the presence of which was observed in the Antarctic using ground based instruments (Kreher et al., 1997) and via satellites (Hollwedel et al., 2004). The presence of iodine oxide (IO) in the Antarctic atmosphere was also confirmed through integrated column measurements from the ground (Frieß et al., 2001). Later, long term ground-based observations of RHS made at Halley Bay showed the critical role that bromine and iodine compounds play in regulating the oxidizing capacity, causing ozone depletion and new particle formation in the Antarctic MBL. These ground-based observations showed that both IO and BrO, are present at elevated concentrations in certain parts of the Antarctic MBL, and show a significant seasonal variation peaking in the spring, with elevated concentrations observed through the summer (Saiz-Lopez et al., 2007a, 2008). Satellite-based observations of both IO and BrO reported a similar annual cycle, although with large geographical differences (Hollwedel et al., 2004; Richter et al., 2002; Saiz-Lopez et al., 2007b; Schönhardt et al., 2008, 2012; Theys et al., 2011; Wagner et al., 2001). These satellite observations have been validated by ground-based observations, although most of them have hitherto focused around the Weddell Sea (Atkinson et al., 2012; Buys et al., 2013; Frieß et al., 2001, 2010; Saiz-Lopez et al., 2007a, 2008). Ground-based observations have also been made at McMurdo Sound, near the Ross Sea, where lower concentrations of IO were observed (Hay, 2010). Additional observations over the 2011-2012 summer were made at Dumont d'Urville using a cavity enhanced absorption spectroscopy based instrument and showed a maximum of 0.15 pptv of IO (Grilli et al., 2013). However, to date observations of IO have not been reported in the Indian Ocean sector of the Antarctic peninsula (Saiz-Lopez et al., 2012; Saiz-Lopez and von Glasow, 2012).





Ground based observations at Halley Bay and in the Weddell Sea suggest that the main source of
iodine compounds is the sea ice region (Atkinson et al., 2012; Saiz-Lopez et al., 2007a). The exact
process is still not known, although a mechanism for biologically-induced iodine emissions from
sea-ice has been suggested based on the idea that micro-algae are the primary source of iodine
emissions in this environment (Saiz-Lopez et al., 2015a). There are further questions regarding the
propagation of reactive iodine chemistry across the continent because satellite observations show
the presence of IO deep within the Antarctic continent, even as far as the South Pole (Saiz-Lopez
et al., 2007b; Schönhardt et al., 2008). However, although enhanced, the observed IO column
densities are close to the detection limit of the satellite instrument and are therefore subject to
uncertainties. One study by Frieß et al. (2010) suggested a strong source within the snowpack,
which hints at active recycling and re-emission of IO aiding the long transport inland. However,
questions remain about why such a source would function only in parts of the continent and why
the primary source is different from the Arctic, where much lower peak concentrations are
sporadically observed (Mahajan et al., 2010; Saiz-Lopez and Blaszczak-boxe, 2016). To further
understand the sources of iodine in the polar environment, understanding the geographical
distribution is critical. Satellite observations play a useful role for this, although validation of the
satellite observations using ground-based instruments is necessary to ascertain their accuracy to
observe IO in the Antarctic troposphere.
Questions also remain about the vertical profiles of iodine compounds in the Antarctic boundary
layer and above the boundary layer. Modelling based studies have suggested a strong gradient
from the surface to the edge of the boundary layer (Saiz-Lopez et al., 2008). Only once in the past
have vertical profiles of IO been measured in Antarctica. These measurements were made at
McMurdo Sound in East Antarctica (Hay, 2010). Observations over two "golden days" in 2006





and 2007 show surface concentrations of about 1 pptv, decreasing to ~0.2 pptv at about 200 m,
before reaching a second maximum of 0.6 pptv at ~700 m. However, models did not reproduce
this profile shape and the authors did mention that the observations had large uncertainties with
the a priori providing most of the information for the profile retrieval (Hay, 2010). In most models,
the assumption is that the source is from the snowpack, with photochemistry in the atmosphere
resulting in a steady decrease with altitude. However, considerable challenges remain in
reproducing the surface variation and vertical gradients in addition to the geographical distribution
(Fernandez et al., 2019). More recent modelling studies combined with aircraft observations
suggest that the gradient is not very sharp all over the globe, with a significant free tropospheric
and stratospheric contribution to the total column (Koenig et al., 2020; Saiz-Lopez et al., 2015b),
although such observations have still not been done in the Antarctic. One of the main reasons for
the uncertainties in models is the lack of consistent measurements of vertical gradients across the
world, especially in the Polar Regions like Antarctica to validate these model simulations.
Considering the uncertainties in the satellite observations and questions regarding the sources and
vertical and geographical distribution of IO, further observations are necessary. Here we present
observations made at two new locations in Antarctica over three summers and compare them with
the satellite-based retrievals and past observations.

**2. Methodology**

Figure 1 shows the location of the two Indian Antarctic stations, Maitri (11.73 °E , 70.77 °S) and
Bharati (76.19 °E, 69.41 °S). The other stations where observations of IO have been reported in
the past are also marked on the map. Observations of IO and the oxygen dimer ($O_4$) were made



using the Multi-Axis Differential Optical Absorption Spectroscopy technique (MAX-DOAS) over
three summers: February-March 2015 as a part of the 34[th] Indian Scientific Expedition to
Antarctica (ISEA-34), November 2015 – February 2016 as a part of ISEA-35 and January-
February 2017 as a part of ISEA-36.
Observations at the Maitri station were made over a short span of 9 days (9[th] March – 17[th] March
2015) and only during ISEA-34. The research station is in the ice-free rocky area on the
Schirmacher Oasis. The MAX-DOAS instrument was installed in a summer-time residential
container, ~150 m north the station, about 120 m above sea level during the ISEA-34. The scanner
unit was mounted on top of the container with the clear line of sight to the horizon. The scanner
pointed ~60.0° with respect to magnetic north. The spectrometer unit was kept inside the container,
which was temperature controlled. The open ocean is 125 km north of Maitri.
Observations at the Bharati station were made for 10 days (9[th] February-18[th] February 2015) during
ISEA-34, for 63 days (30[th] November 2015 – 1[st] February 2016) during ISEA-35 and for 35 days
(5[th] January-11[th] February 2017) during ISEA-36. The station is located between the Thala Fjord
and Quilty Bay, east of the Stornes Peninsula. The MAX-DOAS instrument was installed in a hut
on top of a ridge around 200 m south-west of the Bharati station and was approximately 56 m
above sea level. The scanner unit was mounted on the wall of the hut and had a clear line of sight
to the horizon, pointing -23.2° with respect to the north, overlooking the open ocean. The coastline
is within 500 m of the hut, but it becomes ice free from mid-January to late March. Depending on
the sea ice conditions, the open ocean is within 8-10 km north from the end of November.
The MAX-DOAS instrument (EnviMes) makes use of scattered sunlight along different elevation
angles and by combination of several lines of sight including the zenith. The concentration of an





absorber in the boundary layer can be obtained either in a first approximation by a simple
geometric approach or by simulating the light path with a radiative transfer model taking into
account also multiple scattering effects and the correct treatment of the aerosol loading in the
atmosphere (Hönninger et al., 2004; Platt and Stutz, 2008; Wagner et al., 2004). The instrument
consists of an indoor unit, housing a spectrometer with a spectral resolution of 0.7 nm (UV: 301.20-
463.69), which is connected to an outdoor unit, containing a scanning telescope. Discrete elevation
angles (1°, 2°, 3°, 5°, 7°, 10°, 20°, 40°, and 90°) were recorded for a total exposure time of 1
minute each during all the three campaigns. The spectra were recalibrated before analysis using
mercury emission lines recorded at the end of each day. For DOAS retrieval, the QDOAS 3.2
software was used (Fayt and Van Roozendael, 2013). For estimation of the $O_4$ Differential Slant
Column Densities (DSCDs), the cross-sections of $O_4$ (Thalman and Volkamer, 2013) at 293K;
$NO_2$ (Vandaele et al., 1998) at 294 K and 220 K (orthogonalized to $NO_2$ at 294 K); $O_3$ (Bogumil
et al., 2003) at 223 K and 243 K (orthogonalized to $O_3$ at 243 K); HCHO (Meller and Moortgat,
2000) at 298 K; HONO (Stutz et al., 2000) at 296 K were used in the 351-390 nm window. The
cross-sections used for IO retrieval in the 417-440 nm spectral window were: IO (Gómez Martín
et al., 2005), $NO_2$ 220 K and 298 K (Vandaele et al., 1997), $H_2O$ (Rothman et al., 2013), $O_4$
(Thalman and Volkamer, 2013) and $O_3$ (Bogumil et al., 2003). In addition to these cross-sections
a ring spectrum (Chance and Spurr, 1997), a second ring spectrum following Wagner et al. (2009),
and the 3[rd] order polynomial were used for both windows. The zenith spectrum from each scan
was used as a reference to remove contribution from possible free tropospheric or stratospheric
absorption. An example of a DOAS fit for $O_4$ and IO are given in Figure S1. Surface mixing ratios
and the total vertical column densities (VCDs) were retrieved from the MAX-DOAS DSCDs of
IO and $O_4$ by employing the Mainz Profile Algorithm (MAPA) (Beirle et al., 2018). Only





observations with solar zenith angles (SZA) less than 75° were used for the profile retrievals due
to the large path lengths through the stratosphere for high SZA angles. This algorithm uses a two-
step approach to determine the trace gas vertical profiles. In the first step, the aerosol profiles are
retrieved using the measured $O_4$ DSCDs. A Monte Carlo approach is utilized to identify the best
ensemble of the forward model parameters (column parameters (c) (VCD for trace gases and
aerosol optical depth for aerosol), height parameter (h) and shape parameter (s)), which fit the
measured $O_4$ DSCDs for the sequence of elevation angles. In the second step, the aerosol profiles
retrieved from the $O_4$ inversion are used as an input to retrieve similar model parameters (c, h, and
s) for IO. The state of the atmosphere was calculated using the pressure and temperature profiles
observed by the in situ radiosondes, which were launched once a week at both the stations. An
angstrom exponent of 1 was used for the difference in the retrieval wavelengths as per observations
made at Bharati in the past (Prakash Chaubey et al., 2011). Within MAPA, the differential air mass
factors (AMFs) are calculated offline with the radiative transfer model McArtim (Deutschmann et
al., 2011) for fixed nodes for each parameter and stored as a lookup table (LUT) for quick analysis.
To assess the quality of the retrievals, MAPA also provides "valid", "warning" or "error" flags for
each measurement sequence, which are calculated based on pre-defined thresholds for various fit
parameters. For further details about MAPA, please refer to the description paper by Beirle et al.
(2018). Additionally, MAPA also provides the option to use a scaling factor for significant
mismatch between the modelled and measured $O_4$ DSCDs, which has been shown to be close to
0.8 in the past (Wagner et al., 2019). Using the variable option, where the model estimates the
scaling factor, the estimated value of which ranged between 0.75 and 0.9. Hence a scaling factor
of 0.8 was applied for all the campaigns.





We also make use of the IO vertical column densities retrieved using the SCanning Imaging
Absorption spectroMeter for Atmospheric CHartographY (SCIAMACHY), a UV-vis-NIR
spectrometer onboard the ENVISAT satellite (Burrows et al., 1995). Observations from
SCIAMACHY stopped due to instrumental problems in April 2012. Here we make use of the mean
from 2004-2011 to look at the geographical distribution and compare it with the ground-based
observations made during this study. Further details about the IO retrieval algorithm and the
SCIAMACHY instrumental setup can be found elsewhere (Schönhardt et al., 2008, 2012).

**3. Results and Discussion**
**3.1 Meteorological parameters**
Figure 2 shows the 5-day back-trajectories arriving every hour at the two stations at a height of 10
m on the days that the DOAS measurements were conducted as a part of the three ISEA
expeditions. The back-trajectories were calculated using the HYbrid Single-Particle Lagrangian
Integrated Trajectory (HYSPLIT) using the using the EDAS-40 km database (Draxler and Rolph,
2003). The trajectories show that the airmasses sampled throughout the three expeditions were
from either a remote oceanic region, coastal Antarctica, or the continental shelf. In general, most
of the trajectories show that the airmasses had travelled over hundreds of kilometres over the last
five days. For the local meteorological conditions, Figure 3 (top panels) show the wind direction
at the Bharati station. Most of the time, the wind was from the ocean, with the winds coming from
the north-west sector and a few instances of northern and north eastern winds. This was during all
the three expeditions at the Bharati station. The wind speed was mostly below 20 knots (~10 m s$^-$
$^1$) for all the campaigns, although periods of high winds were observed during ISEA-35 and ISEA-

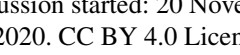



36, which were of a longer duration than ISEA-34. The temperature at the station hovered between
-5°C and +5°C, through the summer period, with higher values closer to noon (Figure 3, middle
panels). The humidity fluctuated from 40% to above 90%. The radiation followed a clear diurnal
pattern, with the highest values seen around local noon and minima at local midnight. Considering
that this region sees light for 24 hours, the radiation also showed a non-zero minima between
November to January (Figure 3, bottom panels). However, in February, a clear night-time is seen
in the radiation data. Finally, a measure of the cloudiness was also tracked using visual full sky
cloud cover observations. Any cloud cover of more than 30% was considered to be cloudy (cloud
flag value of 1), which helps in filtering the MAX-DOAS observations. In addition to the visual
inspection of the sky, which was performed once an hour, a second cloud index was calculated
based on the ratios of the radiances at 320 nm and 440 nm from the 3° and zenith spectra (Mahajan
et al., 2012; Wagner et al., 2014). Both the manual and radiance-based indices showed a close
match, indicating that cloudy conditions were well discerned by the cloud index calculation.
Meteorological data was unfortunately not available at the Maitri station.
**3.2 Differential Slant Column Densities (DSCDs)**
Figure 4 shows the observed $O_4$ DSCDs at different elevation angles for all the campaigns. $O_4$
DSCDs were found to be higher at lower elevation angles, as expected, which is because the $O_4$
concentration is proportional to the square of the oxygen pressure and thus increases towards the
surface. This also suggests that the aerosol loading was low in the atmosphere. Photons travel
longer paths at lower elevation angles and interact more with tropospheric absorbing species before
reaching the instrument resulting in a decreasing profile with increasing elevation angles. The
average residual root mean square (RMS) and $2\sigma$ detection limit for the $O_4$ DSCDs were $4.46 \times 10^{-}$
$^4$ (range: $1.56\text{-}10.01 \times 10^{-4}$) and $2.11 \times 10^{42}$ molecules$^2$ cm$^{-5}$ (range: $0.72\text{-}4.66 \times 10^{42}$ molecules$^2$





cm$^{-5)}$), respectively (Figure 4). The O$_4$ DSCDs were then used to estimate the aerosol profiles and
hence the IO mixing ratios, as described earlier in section 2.
Figure 5 shows the observed IO DSCDs at different elevation angles for all the campaigns. The
IO DSCDs were found to be higher at lower elevation angles, which indicates a decreasing gradient
in the IO vertical profile. The residual RMS was in the 1.15-9.73×10$^{-4}$ range (mean: 3.46×10$^{-4}$),
resulting in 2σ IO DSCD detection limits of 6.57×10$^{12}$ to 5.71×10$^{13}$ molecules cm$^{-2}$ (mean
1.88×10$^{13}$ molecules cm$^{-2}$) (Figure 5). For several days, only the lowermost elevation angles were
found to be above the two-sigma detection limit of the instrument. Higher IO DSCDs were
observed at high SZAs, which is related to an increase in the path length. However, only
observations with SZA<75° were used to estimate the vertical profiles and surface mixing ratios
using the aerosol profiles derived using the O$_4$ DSCDs, as described earlier in section 2. A zoomed
in view of two days for both the O$_4$ and IO DSCDs is shown in Figure S2, which clearly shows
the decreasing gradient with increasing elevation angles.
**3.3 IO Vertical Column Densities (VCDs) and mixing ratio profiles**
The O$_4$ and IO DSCDs were used to retrieve the vertical column densities and the vertical profiles
for aerosols and IO. A comparison of the MAX-DOAS observed O$_4$ DSCDs with the MAPA
modelled DSCDs for all the four campaigns are shown in Figure S3, while Figure S4 shows a
similar plot for the IO DSCDs. Figure 6 and 7 show the MAPA calculated AODs and IO VCDs
for all the campaigns. Several datapoints are flagged as error or warnings, with a few scans giving
a 'valid' flag. In the case of aerosols, the warning or error flags are mainly for scans which were
during cloudy weather (Figure S5 shows the data which were flagged as 'bad' and 'warning' along
with the valid scans). As mentioned above, the cloud cover was regularly measured throughout the





campaigns as a part of the meteorological observations. In addition to visual observations, we also
computed the cloud index following past works based on MAX-DOAS observations (Mahajan et
al., 2012; Wagner et al., 2014), which confirmed that the error and warning flags were during cloud
cover periods. For the valid scans, the aerosol optical depth (AOD) ranged between 0.002 and
0.016, with a mean value of 0.003 for ISEA-34 at Bharati; between 0.001 and 0.067, with a mean
value of 0.011 for ISEA-34 at Maitri; between 0.001 and 1.866, with a mean value of 0.037 for
ISEA-35 at Bharati; and between 0.001 and 0.878, with a mean value of 0.016 for ISEA-36 at
Bharati (Figure 6). The low values are expected considering the pristine conditions in Antarctica,
although during a couple of scans elevated levels were observed as demonstrated by the maximum
value during ISEA-35 and ISEA-36. In the case of IO, there were far fewer valid retrieved profiles
as can be seen in Figure 7 (Figure S6 shows the data which were flagged as 'bad' and 'warning'
along with the valid scans). One of the main reasons is that for most of the scans the IO DSCDs at
higher elevation angles are below the detection limit and hence not enough information is available
for the model to retrieve a valid vertical profile. In the case of IO VCDs, there were only two scans
which showed the valid flag over the 10-day period during the ISEA-34 campaign at Bharati due
to adverse weather conditions leading to mostly cloudy weather. Thus, the VCD value of $2.83\times10^{12}$
molecules cm$^{-2}$, should be treated with some caution. In Maitri during ISEA-34, the IO VCD
ranged between $2.37\times10^{12}$ molecules cm$^{-2}$ and $4.25\times10^{12}$ molecules cm$^{-2}$, with a mean value of
$3.40 \pm 0.57\times10^{12}$ molecules cm$^{-2}$. During ISEA-35 at Bharati, which had the highest number of
valid scans over the four campaigns, the IO VCDs ranged between $0.01\times10^{12}$ and $5.86\times10^{12}$
molecules cm$^{-2}$, with a mean value of $2.62 \pm 1.16\times10^{12}$ molecules cm$^{-2}$. During ISEA-36, the IO
VCDs ranged between $2.78\times10^{12}$ molecules cm$^{-2}$ and $4.90\times10^{12}$ molecules cm$^{-2}$, with a mean value
of $3.92 \pm 0.79\times10^{12}$ molecules cm$^{-2}$ at Bharati.



In addition to the VCDs, vertical profiles of aerosols (Figure S7) and IO were estimated using
MAPA. Figure 8 shows the typical vertical profiles of IO mixing ratios over the four expeditions.
The surface mixing ratios for the valid scans range between 0.2 and 1.3 pptv. The surface
concentrations observed at both Maitri and Bharati are lower than observations in the Weddell Sea
region, where summer time concentrations exceeding 6 pptv have been reported in the past
(Atkinson et al., 2012; Saiz-Lopez et al., 2007a), or at the Neumayer station, where long-term
zenith sky DOAS measurements of IO suggest mixing ratios as high as ~10 pptv during the
summer (Frieß et al., 2001). It should be noted that although elevated concentrations were observed
at Halley, the average summer concentration, measured only 4 m above the snowpack using a
Long-Path DOAS instrument, was about 3 pptv, approximately a factor of three higher than the
observations at Bharati and Maitri. Considering that the MAX-DOAS retrieved profiles are not
very sensitive to the lowermost few meters, this discrepancy is expected. This is because the source
of IO is expected to be from the surface and remote sensing estimates have suggested that high IO
concentrations in the order of 50 ppbv are present in the snow interstitial air (Frieß et al., 2010),
suggesting that snowpack is indeed the source for iodine compounds. If this is indeed the case, a
strong gradient would be observed considering the short lifetime of IO in the atmosphere, and
hence the MAX-DOAS observations would be lower than the LP-DOAS observations. The
observations reported in this study are also similar to measurements at McMurdo Sound, near the
Ross Sea, where MAX-DOAS observations reported a maximum of $2.6 \pm 0.1$ pptv with most of
the observations below 1 pptv during 2006 and 2007 (Hay, 2010). McMurdo Sound is also located
in  the East Antarctic, which shows lower levels of IO in the satellite estimates (Schönhardt et al.,
2008) and in models (Fernandez et al., 2019).





Vertical profiles of IO have been reported only once in the past from Antarctica. These
measurements were made at McMurdo Sound in East Antarctica (Hay, 2010). IO over two days in
2006 and 2007 show typical surface concentrations of ~1 pptv (with a maximum of 2.6 pptv),
decreasing to ~0.2 pptv at about 200 m. A second maximum of 0.6 pptv at ~700 m was also
observed, but the models do not reproduce this profile shape and the observations were subject to
large uncertainties with the vertical profile above 200 m dominated by the a priori (Hay, 2010).
During the four campaigns studied here, elevated concentrations, similar to the surface, were
usually observed until about 400 m. Above this height, there is a decrease, with the retrievals
reducing to below 0.1 pptv (Figure 8). The reducing standard deviations in the profile retrieval
with altitude show that all the profiles which reproduce elevated IO close to the ground approach
zero for higher altitudes, suggesting that most of the IO is within the lower part of the troposphere.
However, this gradient is much more gradual than estimates predicted using the THAMO one-
dimensional model at Halley Bay (Saiz-Lopez et al., 2008). In most models, the assumption is that
the source is from the snowpack, and hence a strong decreasing gradient with altitude has been
predicted (Saiz-Lopez et al., 2008). The gradient of this decrease depends on the photolysis of the
higher oxides, and on the recycling of iodine reservoir species on aerosols, both of which have
uncertainties. When the gradient was estimated in 2008 (Saiz-Lopez et al., 2008), the photolysis
rates for the higher oxides were not available but this has recently been measured in the laboratory
(Lewis et al., 2020) and THAMO needs to be updated accordingly. Another important point to
consider is that the MAX-DOAS observation-based profile retrievals typically get only a couple
of points of information in the boundary layer and are hence not expected to capture this strong
decrease.
**3.4 Comparison with satellite-based estimates**





The satellite-based vertical column densities of IO across the Weddell Sea region, and the region
encompassing Bharati and Maitri are shown in Figure 9. The averaged satellite based VCD
observations suggest that lower levels of IO are expected at both the Indian bases as compared to
places where ground-based observations have been reported in the past, such as Halley Bay and
Neumayer. The averaged value over the eight years of observations at Bharati and Maitri are
between $0.6\text{-}1.4\times10^{12}$ molecules cm$^{-2}$. This is lower than $2.62 \pm 1.16\times10^{12}$ molecules cm$^{-2}$
observed at Bharati during ISEA-35, which was the longest dataset available in this study which
suggests that the ground-based instruments observe larger VCDs as compared to the satellite based
instruments. It should however be noted that the SCIAMACHY data is an average over all the
seasons, and individual daily datapoints as high as $2.1\times10^{12}$ molecules cm$^{-2}$ have been observed.
Figure 10 shows the timeseries for Bharati and Maitri with daily averages (red dots) as well as
monthly averages (blue triangles) for the years 2004 to 2011. Satellite measurements from within
500 km around the stations were included in the analysis.
When the whole IO column is constrained to the lower 400 m, the satellite retrieved VCDs translate
to a range between 0.6 – 1.3 pptv. The daily satellite VCDs tend to exceed these averaged values
and predict mixing ratios as high as 2 pptv. This is similar to the range observed through the three
campaigns reported here, although observations during the spring time, when emissions of iodine
species have been show to peak at Halley Bay (Saiz-Lopez et al., 2007a), were not made over these
three campaigns. During the spring season, values as high as 20 pptv was observed at Halley Bay,
a factor of ten higher than during the summer at the Indian stations. However, the satellite
observations do not show a large peak over the springtime over both Indian stations. Another
outstanding question is whether the satellites are sensitive to the lower 100-200 m, considering the
strong gradient in IO. Figure S8 shows the block AMFs for satellite retrievals showing the



significant difference between the block AMFs over Antarctica at different albedo values. Over
the ice-covered regions in Antarctica, the satellite is sensitive to the lower troposphere as the
albedo is usually 0.9 or above. Observations have shown that open water has an albedo of 0.05–
0.2 (Jin et al., 2004), whereas the albedo of sea ice ranges between 0.6 and 0.7 for bare ice and
0.8–0.9 for snow-covered ice (Perovich et al., 2002). In the case of Bharati, the Quilty Bay is not
ice covered during the summer and hence along the light path in Bharati, the sensitivity of the
satellite is much lower. Use of a higher albedo would result in an underestimation of the VCD by
the satellite, which is the case when compared to the ground-based instruments. At Mairti this
should not be the case considering that Maitri is 125 km inland from the coast, and the ice shelf is
less than 1 km from the station along the light path. It should be noted that the MAPA LUTs are
calculated for a low surface albedo (5%) and hence, at least for some of the measurements, the
surface albedo is probably much larger, especially at Maitri. As far as we understand, the effect of
the surface albedo mainly cancels out in the MAX-DOAS analysis, but it could be one possible
uncertainty on the retrieval results Another reason for the discrepancy between the ground based
and satellite retrieved VCDs could be the overpass time, which was approximately 09:00 am local
time. Although this should not be a large factor during the summer months due to long sunlit hours,
and that the numbers given above were averages through the entire campaign for the ground-based
observations, measurements at Halley Bay have shown a strong diurnal profile peaking at noon
(Saiz-Lopez et al., 2007a). Hence, it is possible that the ground-based observations, which are
filtered for SZA>75°, capture higher values than the satellite.
Finally, a point to consider is that the satellite data available from SCIAMACHY is for the period
of 2004-2011, whereas the MAX-DOAS observations were conducted over three summers from
2015. This temporal discrepancy, although small considering the long satellite dataset, could


contribute to the difference in the retrieved VCDs. Recent observations of iodine in ice-cores in
the Alpine region and over Greenland have shown an increasing trend for atmospheric iodine in
the northern hemisphere (Cuevas et al., 2018; Legrand et al., 2018). In the Antarctic only seasonal
and geographical variations in halogens in the ice have been studied and no long term dataset is
available (Vallelonga et al., 2017). The main cause for this increase is suggested to be an increase
in tropospheric ozone, which drives the emission of iodine compounds from the ocean surface
through heterogenous chemistry at the ocean interstitial surface (Carpenter et al., 2013). Although
questions regarding the strength of this inorganic source in affecting IO concentrations in the
Southern Ocean remain (Inamdar et al., 2020; Mahajan et al., 2019), it is possible that the
discrepancy between the satellite and ground based data is because of a different time period.
However, no increasing trend was observed in the satellite data for the period between 2004-2011
(Figure 10), which suggests that a factor of three increase in the VCDs is most likely due to a
difference in the measurement technique and sensitivities rather than a change in the emissions.

**3.5 Airmass origin dependence**

Year-long observations at Halley Bay in West Antarctica, which were made using the LP-DOAS
instrument, suggested a oceanic primary source (Saiz-Lopez et al., 2007a). This was shown
through the tracking of airmass back-trajectories, which displayed that elevated levels of IO were
present in airmasses that passed over the coastal and oceanic region compared to the airmasses
that had only continental exposures. However, even in airmasses that had passed only over the
continent for the past five days, the IO levels were still above the detection limit, which suggested
that even if the primary source is oceanic, a secondary source from the snow pack contributed to





the atmospheric IO. Indeed, subsequent studies have tried to explain the snowpack source through
recycling of primary emissions from the ocean (Fernandez et al., 2019) and one study has even
suggested a strong snowpack source based on simulated observations (Frieß et al., 2010). Although
the levels of IO are much lower than the peak concentrations seen at Halley Bay, we studied the
back-trajectories to see if the origin of airmasses lead to a difference in the observed IO levels at
both Bharati and Maitri. Considering the short lifetime of reactive iodine compounds in the
atmosphere, we calculated the exposure of each HYSPLIT calculated back-trajectory according to
the region it passed over the last 12 hours. Depending on where the trajectories spend the most
amount of time, they were classified into coastal, continental, and oceanic airmasses. The coastal
region was defined as a 0.5° belt along the Antarctic coastline, with regions to the north and south
of this belt considered to be oceanic and continental (Figure S9). Using the profiles which were
valid, no clear dependence on the airmass origin was observed. Indeed, most of the data points at
both stations corresponded to airmasses which were either coastal or continental (Figure S10) and
is representative of the typical wind patterns during the summer season. Thus, using this dataset,
it was not possible to draw any conclusions regarding the possible sources of IO in this region, and
a longer study is needed in the future.

**4. Conclusions**
This study presents observations of iodine oxide (IO) at the Indian Antarctic bases Maitri and
Bharati made over three summers from 2015 through 2017. IO was observed intermittently during
all the campaigns, with mixing ratios below 2 pptv. Using a profile retrieval algorithm, vertical
gradients of IO were estimated, and these showed a decreasing profile with a peak in the boundary





layer. The vertical columns observed using the ground-based instrument are approximately a factor
of three-five higher than the climatological mean observed by the satellite, which could be due to
a difference in the measurement techniques and sensitivities. Airmass origin analysis using back-
trajectories did not lead to a conclusive answer about the source regions. This study suggests that
a longer dataset over different seasons is necessary to answer the outstanding questions regarding
the sources and seasonal importance of IO in the Indian Ocean sector of Antarctica.

**5. Acknowledgements**
We thank the logistical and scientific teams of the ISEA-34, ISEA-35, and ISEA-36 campaigns
for enabling observations through the expeditions. The ISEA campaigns are organised by the
National Centre for Polar and Ocean Research (NCPOR), Ministry of Earth Sciences (MOES),
Government of India. IITM and NCPOR are funded by MOES, Government of India.
**6. Author contributions:**
ASM conceptualised the research plan and methodology, did the analysis and wrote the
manuscript. MSB did the field observations. SB, TW, NB and ASL helped with the interpretation
of the observations and AS provided the satellite observations and helped interpret them.

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





Figures

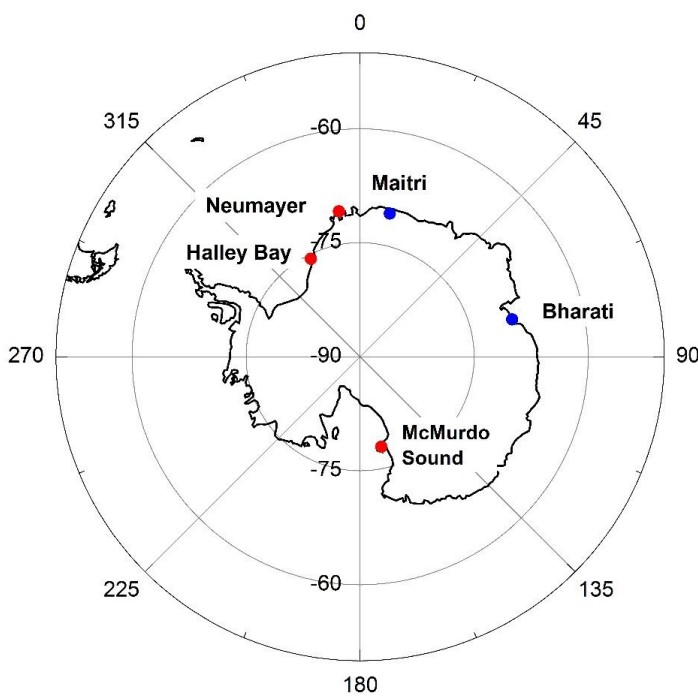


**Figure 1:** Map showing the location of the two Indian Antarctic stations, Maitri and Bharati, where
observations of IO were performed during this study (blue dots). Previous locations that have
reported observations of IO are also marked on the map (red dots).



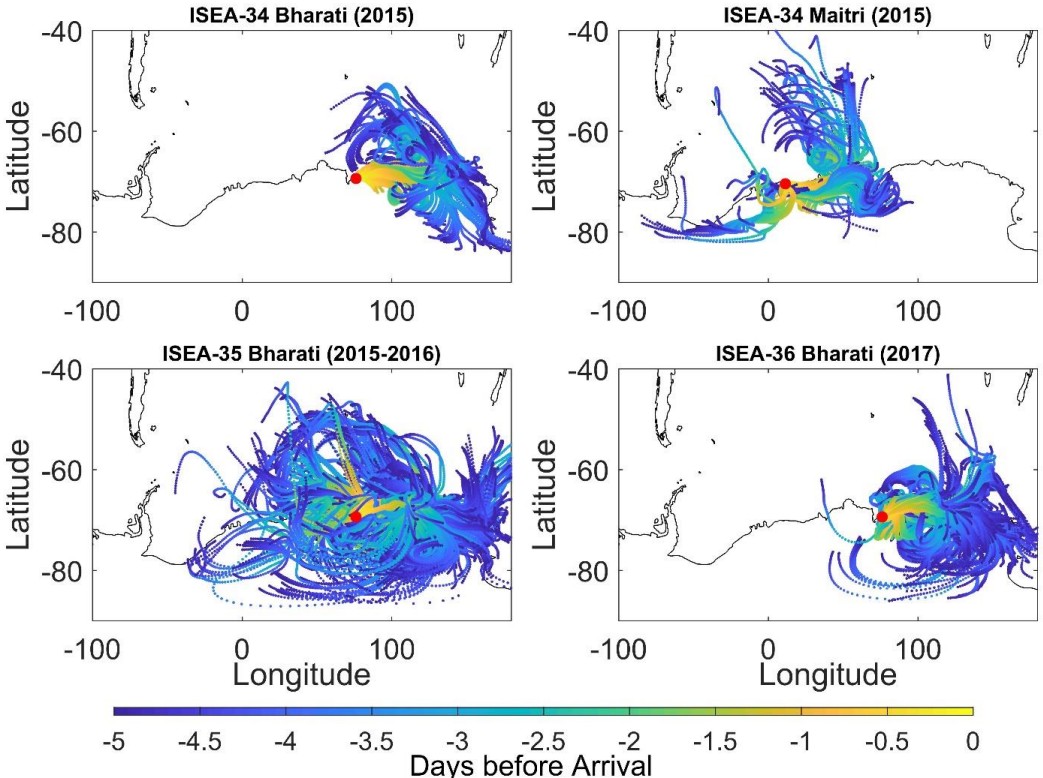


**Figure 2:** 5-day back-trajectories arriving at the two stations on the days that the measurements of

IO were conducted as a part of the 34th, 35th and 36th ISEA expeditions are shown. The back-

trajectories were calculated using the HYbrid Single-Particle Lagrangian Integrated Trajectory

(HYSPLIT) model, arriving every hour (Draxler and Rolph, 2003).

618





619

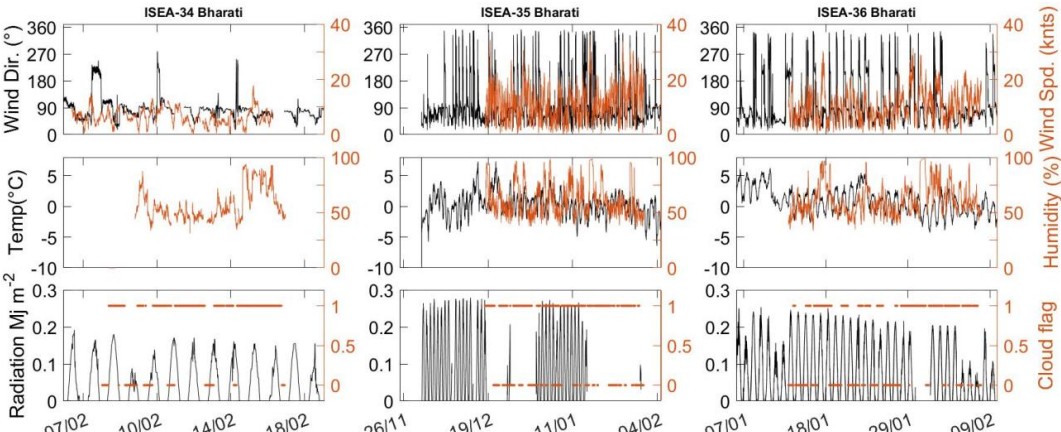

**Figure 3:** Observations of different meteorological parameters that were measured during the various summer campaigns are shown here. The top panels show the wind direction and speed; the middle panels show the temperature and humidity; and the bottom panels show the radiation and cloudiness (1 is defined as 30% cloudy skies and above). Observations of these parameters were not made during the 34[th] ISEA at Maitri and the gaps indicate instrumental or observational issues.

625





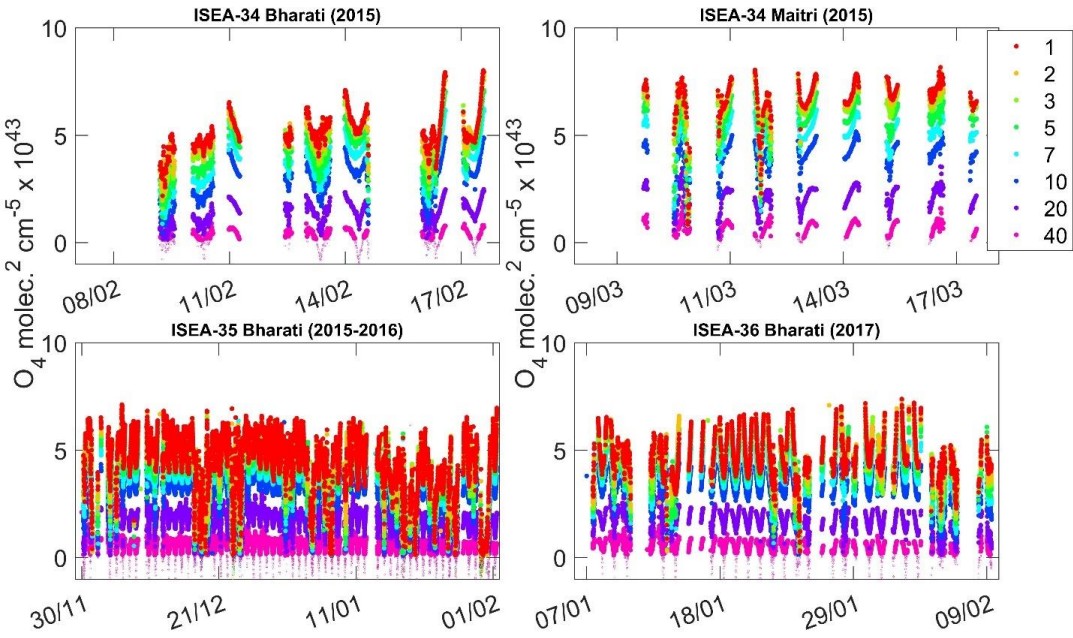

**Figure 4:** O₄ DSCDs observed during the four campaigns are shown. The empty circles represent

values below the 2σ detection limit of the instrument, while the filled circles are values above the

2σ detection limit. The data are color-coded according to elevation angles.





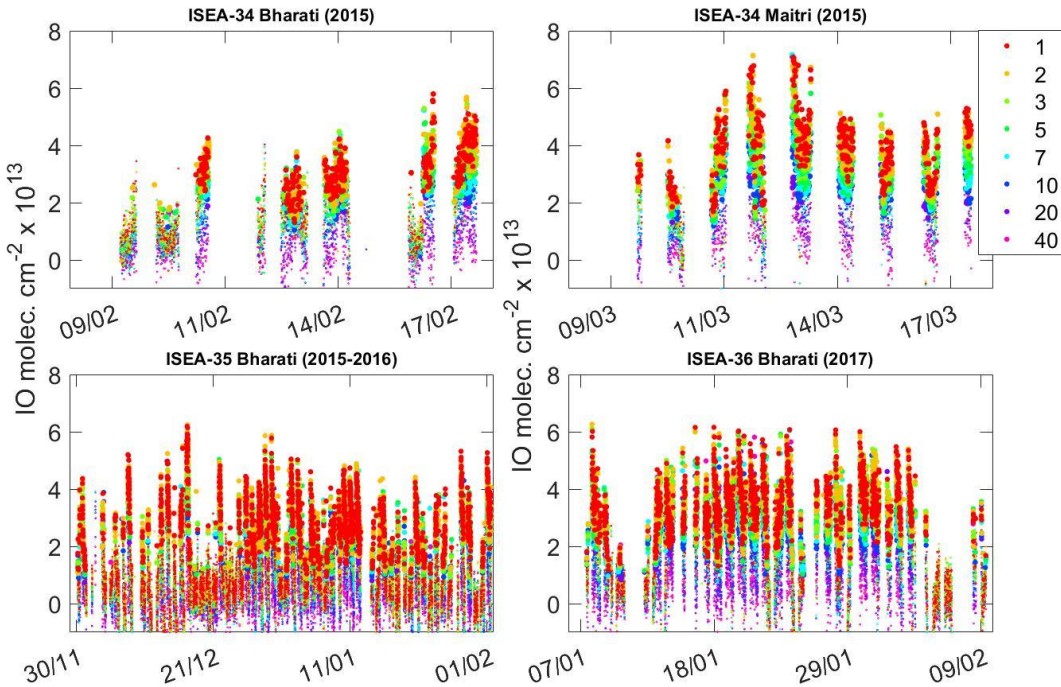


**Figure 5:** IO DSCDs observed during the four campaigns are shown. The smaller circles represent

values below the 2σ detection limit of the instrument, while the bigger circles are values above the
2σ detection limit. The data are color-coded according to elevation angles.






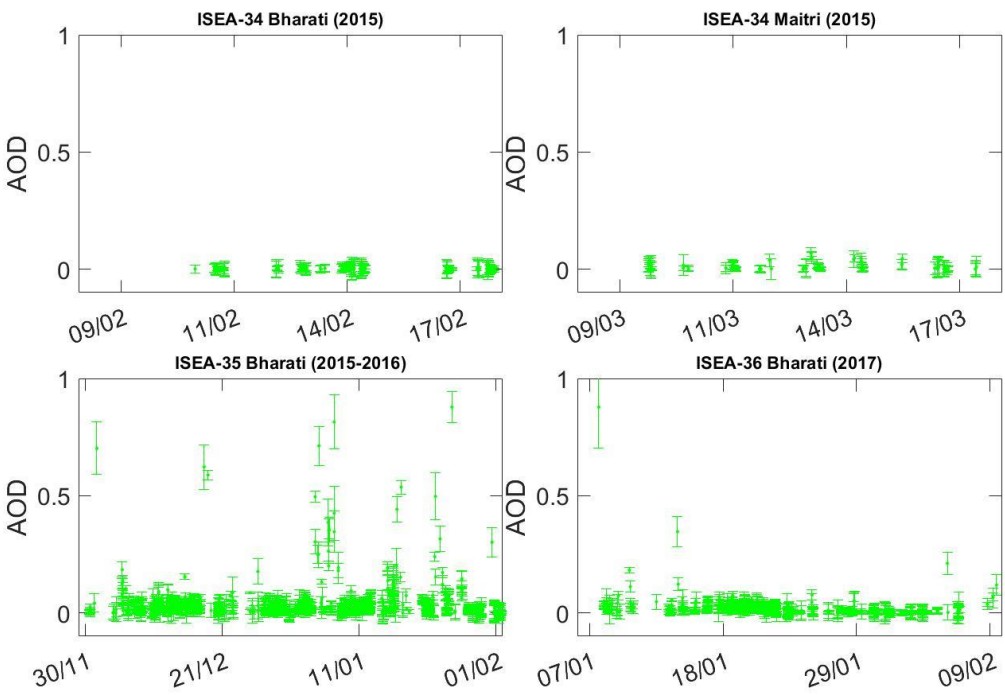


**Figure 6:** AOD timeseries retrieved using the O₄ DSCDs for all the four campaigns are shown.

The data show only the 'good' datapoints, which are reliable and were mostly during clear sky

conditions.







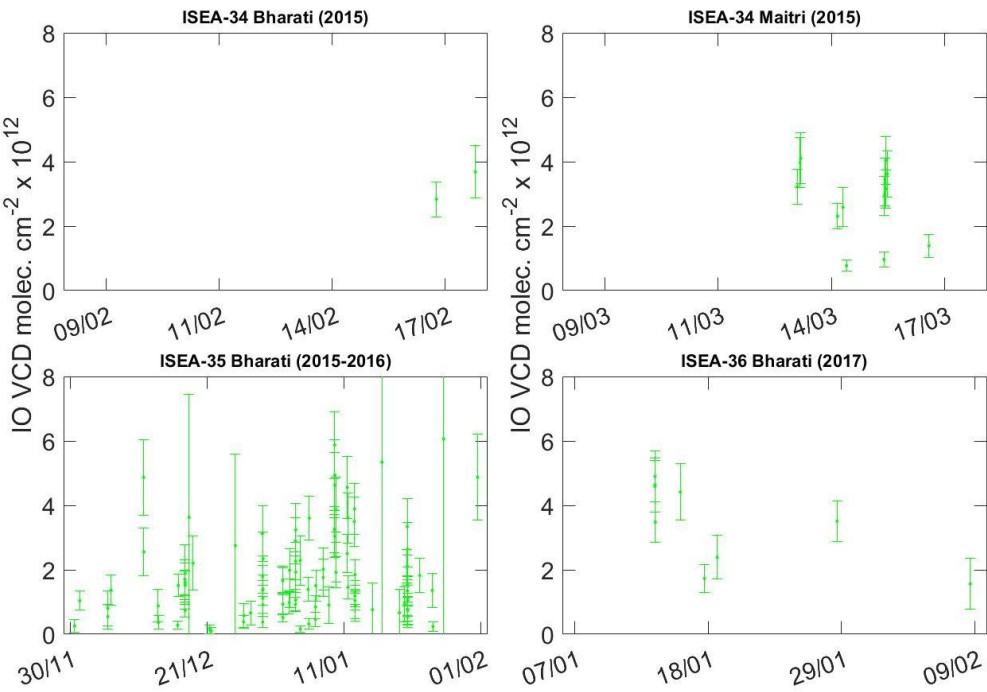

**Figure 7:** Observations of IO vertical column densities observed through all the four campaigns
are shown. These data were mostly during periods of clear sky, and where IO was observed above
the detection limit for most of the set elevation angles, enabling a reliable profile retrieval.





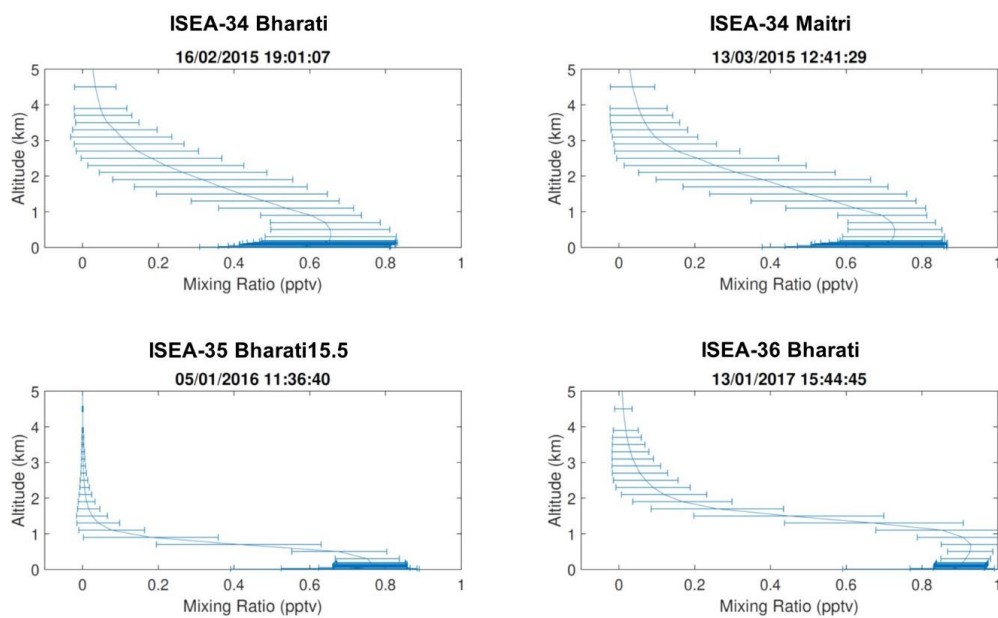


**Figure 8:** Typical examples of IO vertical profiles retrieved during all the four campaigns are

shown.





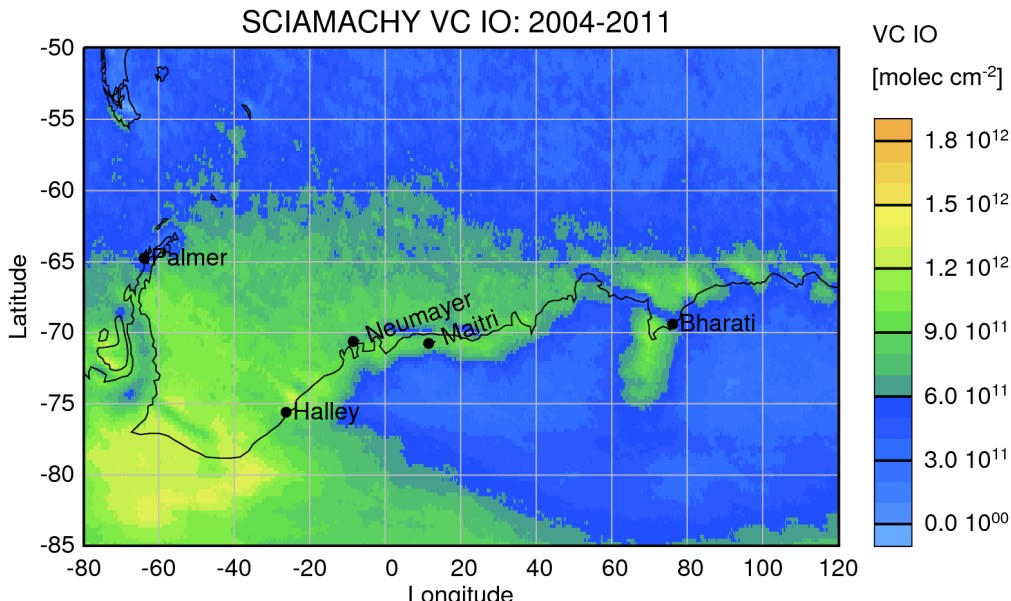


**Figure 9:** Averaged VCDs of IO as retrieved by SCIAMACHY between 2004-2011are shown.

Observations suggest that lower levels of IO are expected at Bharati and Maitri, as compared to

Halley Bay and Neumayer.







**Figure 10:** Timeseries of IO VCD observations at the Bharati station as retrieved by SCIAMACHY. The monthly mean values are shown in blue, and the daily datapoints are shown in red.