# Peer review of "Observations of iodine monoxide over three summers at the Indian Antarctic bases, Bharati"

_Atmospheric Chemistry and Physics, 2020_

## Referee Comment (RC1) · Anonymous Referee #1 · 30 Dec 2020

The paper by Mahajan et al. presents IO observation results measured with a MAX-DOAS at two Indian Antarctic stations (i.e., Bharatic and Maitri) located in East Antarctica. Using a total of 4 data sets from the two stations over the three summers of 2015 – 1017, the authors found that the vertical profiles of IO showed maximum levels near the surface with lower levels at higher altitude. The maximum observed mixing ratio of IO was around 2 ppt and therefore lower than what was measured in previous ground observations in West Antarctica. The IO observations reported in this study were higher than the satellite measurements and the authors proposed that these are likely due to differences in sensitivities. The back trajectory analysis did not show a clear trend in the source of the airmass. Ground observations of halogen gas species

are scarce in polar regions, especially of gas-phase iodine species, and therefore the sources and mechanism driving atmospheric iodine are poorly understood. This study brings in a unique data set which should be publishable after considering major revisions described below.

General comments:

In general, the introduction is lacking some previous studies on reactive halogen species in polar regions. The introduction gives an idea that iodine levels differ geographically, but more details would be helpful. Previous studies that show geographical discrepancies between the Arctic vs Antarctic and East vs West Antarctic should be discussed. For example, similar levels of BrO have been observed between the Arctic and Antarctic while much lower levels of atmospheric iodine have been reported in the Arctic (Hönninger et al., 2004; Pohler et al., 2010; Raso et al., 2017; Schonardt et al., 2008; Tuckermann et al., 1997) compared to the Antarctic. Moreover, both satellite and ground observations show discrepancies between the west and east Antarctica with the Weddell sea being an iodine hotspot. Some relevant references are included in the introduction but describing how previous studies show geographical differences in reactive iodine species would emphasize the uniqueness of the data set presented in the current study, which reports observations in East Antarctica.

For the previous observations presented in the introduction, it would be helpful to include the range or max levels observed and where and which season it was measured for each study. Only parts of the information are included for some studies.

Some background information on the implications of reactive halogen species on ozone depletion events (Barrie et al., 1988; Bottenheim & Gallant, 1986; Kreher et al., 1997; Oltmans & Komhyr, 1986) and new particle formation (Alicke et al., 1999; Allan et al., 2000; Carpenter et al., 1999; C. O'Dowd et al., 1999; C. D. O'Dowd et al., 1998) should be included. Adding background information on how reactive bromine and iodine species go through catalytic cycles that drive ozone depletion events and how the

presence of iodine can accelerate this would be helpful.

In the second paragraph, the authors give examples of the possible source of reactive iodine gas species in Antarctica. It should be clear that previous observations at Halley Bay (Saiz-lopez et al., 2007) and the Weddell sea (Atkinson et al., 2012) proposed sea ice as the source of reactive iodine based on back trajectories and iodocarbon measurements in the sea ice. Additional studies that measured iodocarbons from coastal Antarctica (Carpenter et al., 2007; Fogelqvist & Tanhua, 1995; Reifenhäuser & Heumann, 1992), which can move up the brine channel in sea ice (Garrison & Buck, 1989) and released to the atmosphere, should be discussed and included.

In the third paragraph, discrepancies in the vertical profile of reactive halogens between model simulations and observations are introduced. However, it is not clear throughout the paragraph on specifically 'what' the examples are referring to (e.g., IO, BrO or halogen species in general?). These should be clarified for each of the previous study that'scited.

Figure 8 seems to show one randomly chosen scan of IO vertical profile for each campaign. It is not very clear whether this one selected vertical profile or all the profiles for the whole campaign is what is being compared to previous observations at the Neumayer station and the Weddell sea. This should be clarified. Also, in Figure 8, it would be more helpful to show all the valid vertical profile scans for all four campaigns with a median (or mean) with standard deviations of both the vertical profiles and the surface. The averaged surface IO levels for the whole campaign should be compared to previous observations rather than one randomly chosen scan.

As mentioned in the discussion (Ln 312-313), the satellite observation results were averaged over 2004-2011 and compared to the observations in this study. However, since reactive halogen species have strong seasonal variations, it would be better to average the summertime satellite observations of IO (e.g., November to February) for Figure 9 and when comparing with the ground measurements from Bharati and Maitri.

[Figure]

It should also be noted that uncertainties exist when averaging satellite observed IO concentrations within 500 km range of each station (the distance between the Neumayer station and the Maitri station is only $\sim$ 800 km), especially when there are large variations of IO level within that distance as can be expected from Figure 9.

Specific comments:

Ln 40-42 I assume this is from Saiz-Lopez et al. (2007)? Please include reference. Ln 81 specify which type of model and what type of sources and chemistries were included in the model Ln 84 sources of 'what'? Ln 88 gradient of 'what'? Ln 187 – 189 It seems like during ISEA-34, the winds were mostly E to NE while during ISEA-35 and ISEA-36, the winds varied more. Ln 220 – 221 "For several days,..." specify if this is for all four campaigns or for a specific campaign. Ln 229 – 231 It should be noted that these are examples of one randomly chosen scan for each campaign. Ln 232 – 233 Rather than using 'several' or 'a few' for describing data flagged as 'bad' or 'valid', it would be better to just describe the % (or # of scans) that gave good data that were flagged 'valid'. (same with Ln 246 – 247) Ln 239 – 243 include standard deviation Ln 251 specify what 2.83X1012 molecules cm-2 is. Is this the average? Ln 261 specify altitude of 'surface'. Specify whether 0.2 – 1.3 pptv range of IO is for all the four campaigns throughout the whole observations period with 'valid' scans or for the specific randomly chosen scan for each campaign shown in Figure 8. Ln 309 provide average and standard deviation of satellite observation of IO for each station (i.e., Bharati and Maitri) Ln 384-385 this sentence contradicts with Ln 187 that says most airmasses were from the ocean for all four campaigns.

When describing all the data sets, it's sometimes described as 'three campaigns' (e.g., Line 131, Line 319) or as 'four campaigns' (e.g., in figures captions), which could be confusing. It would be better to be consistent.

There are parts that needs rephrasing. For example: Ln 246 – 247 is same as Ln 234 – 235 Ln 327 – 329 is the same as Figure S8 caption

Figure 1 Add Dumont d'urville station in the figure and include references for each of the locations that previously reported IO.

Figure 2 is difficult to see since there are too many trajectories overlapping. A wider time resolution (e.g., 6 h or 12h) should be sufficient rather than trajectories every hour for the entire three summers.

Figure 3 Add frequency of each data (e.g., 5 min averaged?). It is difficult to interpret the wind directions and speeds during ISEA-35 and ISEA-36 since there are many days with two lines overlapping too much. One option might be smoothing it out by averaging and having one parameter as a marker (e.g., wind speed) and the other color coding (e.g., wind direction) the markers. Same with temperature and humidity, it might be easier to interpret the data if the y-axis scale is adjusted so that the two line (orange and black) don't overlap too much.

Technical corrections:

Ln 40 'year-long' study rather than 'long term'. Ln 139 NO2 at 220 K and 298 K Ln 156 at both stations Ln 164 Additionally, MAPA provides the option Ln 167 the estimated value ranged between Ln 225 of two example days Ln 230 Figure S3, and Figure S4 shows Ln 213- 232 IO VCDs, respectively, for all the campaigns. Ln 338 surface albedo is probably much higher Ln 340 on the retrieval results. (period missing)

Figure S6 data points extend outside the lower limit of y-axis

References

Alicke, B., Hebestreit, K., Stutz, J., & Platt, U. (1999). Iodine oxide in the marine boundary layer. Nature, 397, 572–573.

Allan, B. J., Mcfiggans, G., Plane, J. M. C., & Coe, H. (2000). Observations of iodine monoxide in the remote marine boundary layer. Journal of Geophysical Research, 105(Dll), 14363–14369.

[Figure]

Atkinson, H. M., Huang, R. J., Chance, R., Roscoe, H. K., Hughes, C., Davison, B., Schönhardt, A., Mahajan, A. S., Saiz-Lopez, A., Hoffmann, T., & Liss, P. S. (2012). Iodine emissions from the sea ice of the Weddell Sea. Atmospheric Chemistry and Physics, 12(22), 11229–11244. https://doi.org/10.5194/acp-12-11229-2012

Barrie, L. A., Bottenheim, J. W., Schnell, R. C., Crutzen, P. J., & Rasmussen, R. A. (1988). Ozone destruction and photochemical reactions at polar sunrise in the lower Arctic atmosphere. Nature, 334(14), 6–9.

Bottenheim, J. W., & Gallant, A. G. (1986). Measurements of NOy species and O3 at 82o N Latitude. Geophysical Research Letters, 13(1), 113–116.

Carpenter, L. J., Sturges, W. T., Penkett, S. A., & Liss, P. S. (1999). Short-lived alkyl iodides and bromides at Mace Head, Ireland: Links to biogenic sources and halogen oxide production. Journal of Geophysical Research Atmospheres, 104(D1), 1679–1689. https://doi.org/10.1029/98JD02746

Carpenter, L. J., Wevill, D. J., Palmer, C. J., & Michels, J. (2007). Depth profiles of volatile iodine and bromine-containing halocarbons in coastal Antarctic waters. Marine Chemistry, 103(3–4), 227–236. https://doi.org/10.1016/j.marchem.2006.08.003

Fogelqvist, E., & Tanhua, T. (1995). Iodinated C1-C4 hydrocarbons released from ice algae in Antarctica. In: Grimvall A., de Leer E.W.B.(eds) Naturally-Produced Organohalogens. Envrionment & Chemistry, vol 1. Springer, Dordrecht.

Garrison, D. L., & Buck, K. R. (1989). The biota of Antarctic pack ice in the Weddell sea and Antarctic Peninsula regions. Polar Biology, 10(3), 211–219. https://doi.org/10.1007/BF00238497

Hönninger, G., Leser, H., Sebastián, O., & Platt, U. (2004). Ground-based measurements of halogen oxides at the Hudson Bay by active long-path DOAS and passive MAX-DOAS. Geophysical Research Letters, 31(4), 1–5. https://doi.org/10.1029/2003GL018982

Kreher, K., Johnston, P. V., & Wood, S. W. (1997). Ground-based measurements of tropospheric and stratospheric BrO at Arrival Heights, Antarctica. Geophysical Research Letters, 24(23), 3021–3024. https://doi.org/10.1029/97GL02997

O'Dowd, C. D., Geever, M., Hill, M. K., Smith, M. H., & Jennings, S. G. (1998). New particle formation: Nucleation rates and spatial scales in the clean marine coastal environment. Geophysical Research Letters, 25(10), 1661–1664.

O'Dowd, C., Heard, D. E., McFiggans, G., Creasey, D. J., Smith, M. H., Lee, J. D., Pirjola, L., Pilling, M. J., Hoell, C., Kulmala, M., Plane, J. M. C., & Allan, B. J. (1999). On the photochemical production of new particles in the coastal boundary layer. Geophysical Research Letters, 26(12), 1707–1710. https://doi.org/10.1029/1999GL900335

Oltmans, S. J., & Komhyr, W. D. (1986). Surface ozone distributions and variations from 1973-1984 measurements at the NOAA geophysical monitoring for climatic change baseline observatories. Journal of Geophysical Research, 91(6), 5229–5236. https://doi.org/10.1029/JD091iD04p05229

Pohler, D., Vogel, L., Friess, U., & Platt, U. (2010). Observation of halogen species in the Amundsen Gulf, Arctic, by active long-path differential optical absorption spectroscopy. Proceedings of the National Academy of Sciences, 107(15), 6582–6587. https://doi.org/10.1073/pnas.0912231107

Raso, A. R. W., Custard, K. D., May, N. W., Tanner, D., Newburn, M. K., Walker, L., Moore, R. J., Huey, L. G., Alexander, L., Shepson, P. B., & Pratt, K. A. (2017). Active molecular iodine photochemistry in the Arctic. Proceedings of the National Academy of Sciences, 114(38), 10053–10058. https://doi.org/10.1073/pnas.1702803114

Reifenhäuser, W., & Heumann, K. G. (1992). Determinations of methyl iodide in the Antarctic atmosphere and the south polar sea. Atmospheric Environment Part A, General Topics, 26(16), 2905–2912. https://doi.org/10.1016/0960-1686(92)90282-P

Saiz-lopez, A., Mahajan, A. S., Salmon, R. A., Bauguitte, S. J., Jones, A. E., Roscoe, H.

K., & Plane, J. M. C. (2007). Boundary Layer Halogens in Coastal Antarctica. Science, 348(July), 348–352. https://doi.org/10.1126/science.1141408

Schonardt, A., Richter, A., Wittrock, F., Kirk, H., Oetjen, H., Roscoe, H. K., & Burrows, J. P. (2008). Observations of iodine monoxide columns from satellite. Atmos. Chem. Phys, 8, 637–653.

Tuckermann, M., Ackermann, R., Golz, C., Lorenzen-Schmidt, H., Senne, T., Stutz, J., Trost, B., Unold, W., & Platt, U. (1997). DOAS-observation of halogen radical-catalysed arctic boundary layer ozone destruction during the ARCTOC-campaigns 1995 and 1996 in Ny-Alesund, Spitsbergen. Tellus, 49B, 533–555.

---

## Editor Comment (EC1) · Jayanarayanan Kuttippurath (Editor) · 20 Jan 2021

REVIEW of

**Observations of iodine monoxide over three summers at the Indian Antarctic bases, Bharati and Maitri,** by Mahajan et al.

This MS presents ground-based measurements of Iodine Monoxide in Antarctica and compare that with satellite measurements. The authors also discuss the spatial and temporal variability of IO in terms of measurements at different Antarctic stations in different years or seasons. They have also made an effort to assess their analysis with published results. As these types of measurements are not performed often, and there are large uncertainties and differences between ground-based and satellite measurements, the analysis presented is a valuable addition to existing literature and database. However, some clarifications and a careful revision are needed before it can be recommended for a publication in ACP. Please find the specific comments below.

**Main points:**

1. Please write the significance of IO in the ozone depletion events, and discuss ozone as a source of emissions too. A balanced discussion is needed in this regard to emphasize the importance or relative importance of IO in Antarctica. Please include this in Introduction.

2. Why Neumayer shows 10 pptv, but Maitri 6 pptv? I thought both stations are close enough to show similar IO measurements. You also stated in the discussion that there are no significant inter-annual variability. Please discuss this.

3. You mentioned it is not possible to draw a conclusion about the origin of air masses. What would happen if you make 15-day back trajectory? Will that make any difference in your conclusions? Indeed, you need to consider the lifetime of IO in this case.

4. Write something positive in Conclusions. This is very short and the last sentence is a bit negative.

**Minor points:**

L 23-24: any indications for sources?

L27-28: the high bias is also there for other regions?

L 40: "long-term"

L45, 52: can you please provide a range of values measured

L55: "However, observations of IO have not been reported ….. to date "

L58: ice is everywhere there, please specify the region

L65: please write the detection limit value

L66: The study done by Friess et al…

L75-76: "within and above the boundary layer"

L79: what are these golden days, and why?

L81-83: All measurements and models have problems and uncertainties. Rephrase this sentence. "However, models could not reproduce the measured profiles. There were also uncertainties associated with the *a priori* profiles used in the retrievals." Something like this.

L88: I would use "across the latitudes" than globe

L92: "across the regions, particularly in the Antarctic to validate…"

L167: "Therefore", instead of hence

L174: "during the study period"

L192: -5 °C and +5 °C

L195: the regions has or "experiences"

L202: both manual and radiance-based

L204: Neumayer is nearby; data are available for that station? You could also consider reanalysis data and extract for the station

L233; ", which were"

L236: In line 204, you mentioned that there were no meteorological observations

L248-249: "and hence…." Not needed as you have already stated the reason

L250; "that showed"

L250: "a valid flag"

L252: molec. cm$^{-2}$ and other places too

L265: Neumayer and Maitri are nearby stations. Don't you expect similar range of values at both stations? Is this 4 pptv difference justified?

L270: this is not discrepancy, but difference

L273: two "indeed" there; "snow pack is the primary source for …"

L278: You stated earlier that these measurements were highly weighted by a wrong a priori information?

L285: "models could not"

L296: oxides of what?

L312: ", however, "

L316: satellite measurements within 500 km? Is it comparable?

L319: predict?

L319-322: can you please split the sentence. Hard to comprehend

L322: values …were

L325: this can be another reason for the differences. SCIA measurements are sensitive at these altitudes?

L326: a significant

L368: "regions"

L377: What is the lifetime of IO there? If you consider the short life time and 500 km averaged satellite data, the comparison is fair?

Figure 2: x-axis title reads "Davs" instead of days

Figure 3: The resolution is very poor

Figure 4: the dots of the legend can be little bigger to delineate the colours

---

## Author Comment (AC1) · 11 Mar 2021

**Response to reviewer comments for manuscript number: acp-2020-998**

Comments by reviewers are shown in italic typeface and the responses shown normal typeface.
* * *
Reviewer 1:

*The paper by Mahajan et al. presents IO observation results measured with a MAXDOAS at two Indian Antarctic stations (i.e., Bharatic and Maitri) located in East Antarctica. Using a total of 4 data sets from the two stations over the three summers of 2015 – 1017, the authors found that the vertical profiles of IO showed maximum levels near the surface with lower levels at higher altitude. The maximum observed mixing ratio of IO was around 2 ppt and therefore lower than what was measured in previous ground observations in West Antarctica. The IO observations reported in this study were higher than the satellite measurements and the authors proposed that these are likely due to differences in sensitivities. The back trajectory analysis did not show a clear trend in the source of the airmass. Ground observations of halogen gas species are scarce in polar regions, especially of gas-phase iodine species, and therefore the sources and mechanism driving atmospheric iodine are poorly understood.*

*This study brings in a unique data set which should be publishable after considering major revisions described below.*

RESPONSE: We thank the reviewer for providing detailed constructive comments and suggestions. The following is a point by point response to the review with corresponding changes made to the manuscript. We hope that the manuscript will now be acceptable with these changes.

*General comments:*
*In general, the introduction is lacking some previous studies on reactive halogen species in polar regions. The introduction gives an idea that iodine levels differ geographically, but more details would be helpful. Previous studies that show geographical discrepancies between the Arctic vs Antarctic and East vs West Antarctic should be discussed. For example, similar levels of BrO have been observed between the Arctic and Antarctic while much lower levels of atmospheric iodine have been reported in the Arctic (Hönninger et al., 2004; Pohler et al., 2010; Raso et al., 2017; Schonardt et al., 2008; Tuckermann et al., 1997) compared to the Antarctic. Moreover, both satellite and ground observations show discrepancies between the west and east Antarctica with the Weddell sea being an iodine hotspot. Some relevant references are included in the introduction but describing how previous studies show geographical differences in reactive iodine species would emphasize the uniqueness of the data set presented in the current study, which reports observations in East Antarctica.*

RESPONSE: We have now included a more detailed discussion on the differences as suggested by the reviewer (Line 55).

*For the previous observations presented in the introduction, it would be helpful to include the range or max levels observed and where and which season it was measured for each study. Only parts of the information are included for some studies. Some background information on the implications of reactive halogen species on ozone depletion events (Barrie et al., 1988; Bottenheim & Gallant, 1986; Kreher et al., 1997; Oltmans & Komhyr, 1986) and new particle formation (Alicke et al., 1999; Allan et al., 2000; Carpenter et al., 1999; C. O'Dowd et al.,*

*1999; C. D. O'Dowd et al., 1998) should be included. Adding background information on how reactive bromine and iodine species go through catalytic cycles that drive ozone depletion events and how the presence of iodine can accelerate this would be helpful.*

RESPONSE: The requested details have now been included (Line 35).

*In the second paragraph, the authors give examples of the possible source of reactive iodine gas species in Antarctica. It should be clear that previous observations at Halley Bay (Saiz-lopez et al., 2007) and the Weddell sea (Atkinson et al., 2012) proposed sea ice as the source of reactive iodine based on back trajectories and iodocarbon measurements in the sea ice. Additional studies that measured iodocarbons from coastal Antarctica (Carpenter et al., 2007; Fogelqvist & Tanhua, 1995; Reifenhäuser & Heumann, 1992), which can move up the brine channel in sea ice (Garrison & Buck, 1989) and released to the atmosphere, should be discussed and included.*

RESPONSE: These details are now included (Line 68).

In the third paragraph, discrepancies in the vertical profile of reactive halogens between model simulations and observations are introduced. However, it is not clear throughout the paragraph on specifically 'what' the examples are referring to (e.g., IO, BrO or halogen species in general?). These should be clarified for each of the previous study that's cited.

RESPONSE: Added.

*Figure 8 seems to show one randomly chosen scan of IO vertical profile for each campaign. It is not very clear whether this one selected vertical profile or all the profiles for the whole campaign is what is being compared to previous observations at the Neumayer station and the Weddell sea. This should be clarified. Also, in Figure 8, it would be more helpful to show all the valid vertical profile scans for all four campaigns with a median (or mean) with standard deviations of both the vertical profiles and the surface.*

RESPONSE: Figure 8 shows the typical valid profile from each campaign. The reason we have not included a campaign mean is to show the uncertainty on the individual profiles. We feel that this is more important and useful considering the small number of valid scans, and larger uncertainties in the profile retrieval.

*The averaged surface IO levels for the whole campaign should be compared to previous observations rather than one randomly chosen scan. As mentioned in the discussion (Ln 312-313), the satellite observation results were averaged over 2004-2011 and compared to the observations in this study. However, since reactive halogen species have strong seasonal variations, it would be better to average the summertime satellite observations of IO (e.g., November to February) for Figure 9 and when comparing with the ground measurements from Bharati and Maitri. It should also be noted that uncertainties exist when averaging satellite observed IO concentrations within 500 km range of each station (the distance between the Neumayer station and the Maitri station is only _ 800 km), especially when there are large variations of IO level within that distance as can be expected from Figure 9.*

RESPONSE: The values compared in the section 3.4 are indeed averages. This is now made clear. A brief discussion on the uncertainties due to spatial averaging in now included (Line 342).

Specific comments:

*Ln 40-42 I assume this is from Saiz-Lopez et al. (2007)? Please include reference.*
RESPONSE: Added.

*Ln 81 specify which type of model and what type of sources and chemistries were included in the model*
RESPONSE: Added.

*Ln 84 sources of 'what'?*
RESPONSE: Added.

*Ln 88 gradient of 'what'?*
RESPONSE: Added.

*Ln 187 – 189 It seems like during ISEA-34, the winds were mostly E to NE while during ISEA-35 and ISEA-36, the winds varied more.*
RESPONSE: Corrected.

*Ln 220 – 221 "For several days,: : :" specify if this is for all four campaigns or for a specific campaign. Ln 229 – 231 It should be noted that these are examples of one randomly chosen scan for each campaign.*
RESPONSE: This sentence is about the DSCDs, for which we have shown the full campaign data.

*Ln 232 – 233 Rather than using 'several' or 'a few' for describing data flagged as 'bad' or 'valid', it would be better to just describe the % (or # of scans) that gave good data that were flagged 'valid'.*
RESPONSE: Added (265)

*Ln 239 – 243 include standard deviation*
RESPONSE: Included.

*Ln 251 specify what 2.83X10$^{12}$ molecules cm-2 is. Is this the average?*
RESPONSE: Yes, this is now made clear.

*Ln 261 specify altitude of 'surface'.*
RESPONSE: Added.

*Specify whether 0.2 – 1.3 pptv range of IO is for all the four campaigns throughout the whole observations period with 'valid' scans or for the specific randomly chosen scan for each campaign shown in Figure 8.*
RESPONSE: Now made clear.

*Ln 309 provide average and standard deviation of satellite observation of IO for each station (i.e., Bharati and Maitri)*
RESPONSE: Added.

*Ln 384-385 this sentence contradicts with Ln 187 that says most airmasses were from the ocean for all four campaigns.*

RESPONSE: This is now corrected.

*When describing all the data sets, it's sometimes described as 'three campaigns' (e.g., Line 131, Line 319) or as 'four campaigns' (e.g., in figures captions), which could be confusing. It would be better to be consistent.*
RESPONSE: Corrected.

*There are parts that needs rephrasing. For example: Ln 246 – 247 is same as Ln 234 – 235 Ln 327 – 329 is the same as Figure S8 caption*
RESPONSE: Corrected.

*Figure 1 Add Dumont d'urville station in the figure and include references for each of the locations that previously reported IO.*
RESPONSE: Added.

*Figure 2 is difficult to see since there are too many trajectories overlapping. A wider time resolution (e.g., 6 h or 12h) should be sufficient rather than trajectories every hour for the entire three summers.*
RESPONSE: We agree – however since the back-trajectory analysis has shown that the trajectories do not play a major role, we have decided to keep the figure as it is.

*Figure 3 Add frequency of each data (e.g., 5 min averaged?). It is difficult to interpret the wind directions and speeds during ISEA-35 and ISEA-36 since there are many days with two lines overlapping too much. One option might be smoothing it out by averaging and having one parameter as a marker (e.g., wind speed) and the other color coding (e.g., wind direction) the markers. Same with temperature and humidity, it might be easier to interpret the data if the y-axis scale is adjusted so that the two line (orange and black) don't overlap too much.*
RESPONSE: Added the frequency. Considering the meteorological data is to give an idea about the conditions, we feel the current format is clear. Smoothing the data could cause false interpretation.

*Technical corrections:*
*Ln 40 'year-long' study rather than 'long term'.*
RESPONSE: Corrected.

*Ln 139 NO2 at 220 K and 298 K*
RESPONSE: Corrected.

*Ln 156 at both stations*
RESPONSE: Corrected.

*Ln 164 Additionally, MAPA provides the option*
RESPONSE: Corrected

*Ln 167 the estimated value ranged between*
RESPONSE: Corrected.

*Ln 225 of two example days*
RESPONSE: Corrected.

*Ln 230 Figure S3, and Figure S4 shows*
RESPONSE: Corrected.

*Ln 213- 232 IO VCDs, respectively, for all the campaigns.*
RESPONSE: Corrected.

*Ln 338 surface albedo is probably much higher*
RESPONSE: Corrected.

*Ln 340 on the retrieval results. (period missing)*
RESPONSE: Corrected.

*Figure S6 data points extend outside the lower limit of y-axis*
RESPONSE: Yes, this was done on purpose to make sure that the scale was readable and the points higher than the scale were not valid in any case.

*References*
*Alicke, B., Hebestreit, K., Stutz, J., & Platt, U. (1999). Iodine oxide in the marine boundary layer. Nature, 397, 572–573.*
*Allan, B. J., Mcfiggans, G., Plane, J. M. C., & Coe, H. (2000). Observations of iodine monoxide in the remote marine boundary layer. Journal of Geophysical Research, 105(Dll), 14363–14369.*
*Atkinson, H. M., Huang, R. J., Chance, R., Roscoe, H. K., Hughes, C., Davison, B.,Schönhardt, A., Mahajan, A. S., Saiz-Lopez, A., Hoffmann, T., & Liss, P. S. (2012). Iodine emissions from the sea ice of the Weddell Sea. Atmospheric Chemistry and Physics, 12(22), 11229–11244. https://doi.org/10.5194/acp-12-11229-2012*
*Barrie, L. A., Bottenheim, J. W., Schnell, R. C., Crutzen, P. J., & Rasmussen, R. A. (1988). Ozone destruction and photochemical reactions at polar sunrise in the lower Arctic atmosphere. Nature, 334(14), 6–9.*
*Bottenheim, J. W., & Gallant, A. G. (1986). Measurements of NOy species and O3 at 82o N Latitude. Geophysical Research Letters, 13(1), 113–116.*
*Carpenter, L. J., Sturges, W. T., Penkett, S. A., & Liss, P. S. (1999). Short-lived alkyl iodides and bromides at Mace Head, Ireland: Links to biogenic sources and halogen oxide production. Journal of Geophysical Research Atmospheres, 104(D1), 1679–1689. https://doi.org/10.1029/98JD02746*
*Carpenter, L. J., Wevill, D. J., Palmer, C. J., & Michels, J. (2007). Depth profiles of volatile iodine and bromine-containing halocarbons in coastal Antarctic waters. Marine Chemistry, 103(3–4), 227–236. https://doi.org/10.1016/j.marchem.2006.08.003*
*Fogelqvist, E., & Tanhua, T. (1995). Iodinated C1-C4 hydrocarbons released from ice algae in Antarctica. In: Grimvall A., de Leer E.W.B.(eds) Naturally-Produced Organohalogens. Envrionment & Chemistry, vol 1. Springer, Dordrecht.*
*Garrison, D. L., & Buck, K. R. (1989). The biota of Antarctic pack ice in the Weddell sea and Antarctic Peninsula regions. Polar Biology, 10(3), 211–219. https://doi.org/10.1007/BF00238497*
*Hönninger, G., Leser, H., Sebastián, O., & Platt, U. (2004). Groundbased measurements of halogen oxides at the Hudson Bay by active longpath DOAS and passive MAX-DOAS. Geophysical Research Letters, 31(4), 1–5. https://doi.org/10.1029/2003GL018982*
*Kreher, K., Johnston, P. V., &Wood, S.W. (1997). Ground-based measurements of tropospheric and stratospheric BrO at Arrival Heights, Antarctica. Geophysical Research Letters, 24(23), 3021–3024. https://doi.org/10.1029/97GL02997*

O'Dowd, C. D., Geever, M., Hill, M. K., Smith, M. H., & Jennings, S. G. (1998). New particle formation: Nucleation rates and spatial scales in the clean marine coastal environment. Geophysical Research Letters, 25(10), 1661–1664.

O'Dowd, C., Heard, D. E., McFiggans, G., Creasey, D. J., Smith, M. H., Lee, J. D., Pirjola, L., Pilling, M. J., Hoell, C., Kulmala, M., Plane, J. M. C., & Allan, B. J. (1999). On the photochemical production of new particles in the coastal boundary layer. Geophysical Research Letters, 26(12), 1707–1710. https://doi.org/10.1029/1999GL900335

Oltmans, S. J., & Komhyr, W. D. (1986). Surface ozone distributions and variations from 1973-1984 measurements at the NOAA geophysical monitoring for climatic change baseline observatories. Journal of Geophysical Research, 91(6), 5229–5236. https://doi.org/10.1029/JD091iD04p05229

Pohler, D., Vogel, L., Friess, U., & Platt, U. (2010). Observation of halogen species in the Amundsen Gulf, Arctic, by active long-path differential optical absorption spectroscopy. Proceedings of the National Academy of Sciences, 107(15), 6582–6587. https://doi.org/10.1073/pnas.0912231107

Raso, A. R. W., Custard, K. D., May, N. W., Tanner, D., Newburn, M. K., Walker, L., Moore, R. J., Huey, L. G., Alexander, L., Shepson, P. B., & Pratt, K. A. (2017). Active molecular iodine photochemistry in the Arctic. Proceedings of the National Academy of Sciences, 114(38), 10053–10058. https://doi.org/10.1073/pnas.1702803114

Reifenhäuser, W., & Heumann, K. G. (1992). Determinations of methyl iodide in the Antarctic atmosphere and the south polar sea. Atmospheric Environment Part A, General Topics, 26(16), 2905–2912. https://doi.org/10.1016/0960-1686(92)90282-P

Saiz-lopez, A., Mahajan, A. S., Salmon, R. A., Bauguitte, S. J., Jones, A. E., Roscoe, H. K., & Plane, J. M. C. (2007). Boundary Layer Halogens in Coastal Antarctica. Science, 348(July), 348–352. https://doi.org/10.1126/science.1141408

Schonardt, A., Richter, A., Wittrock, F., Kirk, H., Oetjen, H., Roscoe, H. K., & Burrows, J. P. (2008). Observations of iodine monoxide columns from satellite. Atmos. Chem. Phys, 8, 637–653.

Tuckermann, M., Ackermann, R., Golz, C., Lorenzen-Schmidt, H., Senne, T., Stutz, J., Trost, B., Unold, W., & Platt, U. (1997). DOAS-observation of halogen radical-catalysed arctic boundary layer ozone destruction during the ARCTOC-campaigns 1995 and 1996 in Ny-Alesund, Spitsbergen. Tellus, 49B, 533–555.
* * *
Reviewer 2:

Review of Observations of iodine monoxide over three summers at the Indian Antarctic bases, Bharati and Maitri, by Mahajan et al.

This MS presents ground-based measurements of Iodine Monoxide in Antarctica and compare that with satellite measurements. The authors also discuss the spatial and temporal variability of IO in terms of measurements at different Antarctic stations in different years or seasons. They have also made an effort to assess their analysis with published results. As these types of measurements are not performed often, and there are large uncertainties and differences between ground-based and satellite measurements, the analysis presented is a valuable addition to existing literature and database. However, some clarifications and a careful revision are needed before it can be recommended for a publication in ACP. Please find the specific comments below.

RESPONSE: We thank the reviewer for the positive comments and have changed the manuscript according to the specific comments, responses to which are detailed below.

Main points:

*1. Please write the significance of IO in the ozone depletion events, and discuss ozone as a source of emissions too. A balanced discussion is needed in this regard to emphasize the importance or relative importance of IO in Antarctica. Please include this in Introduction.*

RESPONSE: Added (Line 35).

*2. Why Neumayer shows 10 pptv, but Maitri 6 pptv? I thought both stations are close enough to show similar IO measurements. You also stated in the discussion that there are no significant inter-annual variability. Please discuss this.*

RESPONSE: Maitri sees even lower levels of about 1.3 pptv, while the 6 pptv number is for Halley in summer. However, we still do not understand why such a difference is seen – this also relates to what the drivers of iodine compounds are, which is still not known. We have added this clearly in the modified manuscript (Line 294).

*3. You mentioned it is not possible to draw a conclusion about the origin of air masses. What would happen if you make 15-day back trajectory? Will that make any difference in your conclusions? Indeed, you need to consider the lifetime of IO in this case.*

RESPONSE: Unfortunately, not. First, the 15-day trajectories in the polar region have large uncertainties. Secondly, iodine has a short lifetime in the atmosphere, hence the sources are most likely local and hence studying 15-day trajectories would not give us any more information.

*4. Write something positive in Conclusions. This is very short and the last sentence is a bit negative.*

RESPONSE: We feel that the conclusion is not negative, but rather a call for action from the community to answer the outstanding questions. However, we have added another conclusion, which was that the observations confirmed past hypothesis of a source from the ground considering a sharp gradient (Line 422 and Line 427).

*Minor points:*

*L 23-24: any indications for sources?*
RESPONSE: It is hard to speculate which – this is now added.

*L27-28: the high bias is also there for other regions?*
RESPONSE: Only other observations we have in the past are from the Weddell Sea, which are closer to the satellite estimates.

*L 40: "long-term"*
RESPONSE: Corrected.

*L45, 52: can you please provide a range of values measured*

RESPONSE: Added.

*L55: "However, observations of IO have not been reported ..... to date "*
RESPONSE: Corrected.

*L58: ice is everywhere there, please specify the region*
RESPONSE: Weddell Sea – it is already mentioned in that sentence.

*L65: please write the detection limit value*
RESPONSE: Added.

*L66: The study done by Friess et al…*
RESPONSE: Corrected

*L75-76: "within and above the boundary layer"*
RESPONSE: Corrected.

*L79: what are these golden days, and why?*
RESPONSE: These were days when the instrument was able to retrieve the profiles in their study, and was termed by Hay et al. We have used their terminology when referring to their data.

*L81-83: All measurements and models have problems and uncertainties. Rephrase this sentence. "However, models could not reproduce the measured profiles. There were also uncertainties associated with the a priori profiles used in the retrievals." Something like this.*
RESPONSE: Corrected.

*L88: I would use "across the latitudes" than globe*
RESPONSE: Corrected.

*L92: "across the regions, particularly in the Antarctic to validate…"*
RESPONSE: We removed Antarctica as this is true for the Arctic too.

*L167: "Therefore", instead of hence*
RESPONSE: Corrected.

*L174: "during the study period"*
RESPONSE: Corrected.

*L192: -5 °C and +5 °C*
RESPONSE: Corrected.

*L195: the regions has or "experiences"*
RESPONSE: Corrected.

*L202: both manual and radiance-based*
RESPONSE: Corrected.

*L204: Neumayer is nearby; data are available for that station? You could also consider reanalysis data and extract for the station*

RESPONSE: Since the meteorological factors do not contribute to the conclusions of this study, we do not feel the need to compare data using different sources.

*L233; ", which were"*
RESPONSE: Corrected.

*L236: In line 204, you mentioned that there were no meteorological observations*
RESPONSE: The cloud index was also computed using the radiances of the MAX-DOAS.

*L248-249: "and hence…." Not needed as you have already stated the reason*
RESPONSE: Corrected

*L250; "that showed"*
RESPONSE: Corrected.

*L250: "a valid flag"*
RESPONSE: corrected

*L252: molec. cm-2 and other places too*
RESPONSE: We prefer not using 'molec.' instead of 'molecules'. Both are accepted forms.

*L265: Neumayer and Maitri are nearby stations. Don't you expect similar range of values at both stations? Is this 4 pptv difference justified?*
RESPONSE: We do not understand why we see such a difference. Indeed, it is one of the outstanding questions as mentioned in the response to reviewer 1. We have added this explicitly in the manuscript (Line 294).

*L270: this is not discrepancy, but difference*
RESPONSE: Corrected.

*L273: two "indeed" there; "snow pack is the primary source for …"*
RESPONSE: Corrected.

*L278: You stated earlier that these measurements were highly weighted by a wrong a priori information?*
RESPONSE: Yes, but not for the surface values where more information was available.

*L285: "models could not"*
RESPONSE: The models still do not.

*L296: oxides of what?*
RESPONSE: Added.

*L312: ", however, "*
RESPONSE: corrected.

*L316: satellite measurements within 500 km? Is it comparable?*
RESPONSE: Considering the low values, we need to average larger spatial areas to increase the signal to noise. But we have added a caveat in the manuscript to make this clear (Line 341).

*L319: predict?*
RESPONSE: Corrected.

*L319-322: can you please split the sentence. Hard to comprehend*
RESPONSE: Corrected.

*L322: values ...were*
RESPONSE: Corrected in the earlier comment.

*L325: this can be another reason for the differences. SCIA measurements are sensitive at these altitudes?*
RESPONSE: The block AMFs show the sensitivity and we have discussed this in the next few sentences.

*L326: a significant*
RESPONSE: Corrected.

*L368: "regions"*
RESPONSE: Corrected

*L377: What is the lifetime of IO there? If you consider the short life time and 500 km averaged satellite data, the comparison is fair?*
RESPONSE: This is a difficult question to answer. The atmospheric lifetime is short, however, could be extended due to recycling on aerosols and the snowpack. We still do not understand the full chemistry.

*Figure 2: x-axis title reads "Davs" instead of days*
RESPONSE: Corrected.

*Figure 3: The resolution is very poor*
RESPONSE: High resolution images will be provided, right now the resolution is set by word.

*Figure 4: the dots of the legend can be little bigger to delineate the colours*
RESPONSE: We had tried this; however, bigger dots overlap and reduce the ability to delineate the colours.

---

## Referee Report (RR1)

**Review of "Observations of iodine monoxide over three summers at the Indian Antarctic bases, Bharati and Maitri" by Mahajan et al.**

Mahajan et al present a revised version of their manuscript on IO observations at two Indian stations located in East Antarctica over three summers. The manuscript has improved with additional literature and clarifications of the data. I have remaining questions regarding my previous review and additional suggestions. The original reviewer comment is in *italic grey* and the new comments are in ***blue***.

*Figure 8 seems to show one randomly chosen scan of IO vertical profile for each campaign. It is not very clear whether this one selected vertical profile or all the profiles for the whole campaign is what is being compared to previous observations at the Neumayer station and the Weddell sea. This should be clarified. Also, in Figure 8, it would be more helpful to show all the valid vertical profile scans for all four campaigns with a median (or mean) with standard deviations of both the vertical profiles and the surface.*

> If possible, it is encouraged to include a campaign average and stdev of the valid scans in the supporting information. It would be helpful to include this to understand the general range of the profiles during the campaign. In Ln 309-311, you mention that maximum IO mixing ratios were observed near 400m with a decrease above this. Does 400m correspond to a typical boundary layer height in this region? If so, please include this in the discussion.

> In Ln 315-317, you mention that most models predict a strong IO gradient as the main source is from the surface within the model. Also, in Ln 422-423, a conclusion of "The vertical profiles confirmed past hypothesis of a source from the ground considering a sharp gradient" is made. However, Figure 8 seems like the IO is actually decreasing below ~ 400m towards the surface. Was this trend also observed in other valid vertical profile measurements of IO? It would be interesting to include a discussion of the possible explanations of the decrease in observed IO levels below the boundary layer towards the surface. As it is difficult to see the vertical profiles of IO close to the surface, it is suggested to include a graph similar to Figure 8 but with altitude < 1km in the supporting information.

*Figure 1 Add Dumont d'urville station in the figure and include references for each of the locations that previously reported IO.*

> The references for each of the locations are still not added. Please add these in the figure caption.

*Figure 2 is difficult to see since there are too many trajectories overlapping. A wider time resolution (e.g., 6 h or 12h) should be sufficient rather than trajectories every hour for the entire three summers.*

> Based on the back trajectories in Figure 2, I'm surprised that there's very few airmasses (red dots in Fig. S10) that are categorized 'ocean airmass' as described in Ln 407-413. For example,

during ISEA-35 at Bharati, airmasses are coming from all directions including the ocean. Could you add more details on how the categorization was made based on the back trajectories? In Fig. S10, as it is difficult to see which observation profile corresponds to each type of airmass, it is suggested to just add the airmass category (red dots) for the valid profiles.

*Figure 3 Add frequency of each data (e.g., 5 min averaged?). It is difficult to interpret the wind directions and speeds during ISEA-35 and ISEA-36 since there are many days with two lines overlapping too much. One option might be smoothing it out by averaging and having one parameter as a marker (e.g., wind speed) and the other color coding (e.g., wind direction) the markers. Same with temperature and humidity, it might be easier to interpret the data if the y-axis scale is adjusted so that the two line (orange and black) don't overlap too much.*

Then as the graph is difficult to see, it is recommended to add another row of graphs and separate wind speed and wind direction. Also, please Include time zone in the x axis (or top of each graph) for all the figures including figures 3, 4,5,6,7,8, S2, S3, S4, S5,S6,S7.

**Additional Specific Comments:**
(Ln 70) Please specify what the measured iodocarbons weren't able to explain high levels of. Was it IO/$I_2$?
(Ln 94) Please specify what measurement technique was used for the vertical profile of IO.
(Ln 96) Please specify what the detection limit and uncertainty of IO was for the cited study.
(Ln 164) suggested to use 'large' SZA rather than 'high' SZA throughout the manuscript.
(Ln 183) change to "..the estimated value ranged between.."
(Ln 198) "using the" is used twice
(Ln 279) Just to clarify, the range is for <30m? Please include the average and standard deviation.
(Ln 348) " .. shown to peak.."

---

## Author Response (AR2)

**Response to reviewer comments for manuscript number: acp-2020-998**

Comments by reviewers are shown in italic typeface and the responses shown normal typeface.
* * *
Reviewer 1:

***Review of "Observations of iodine monoxide over three summers at the Indian Antarctic bases, Bharati and Maitri" by Mahajan et al.***

*Mahajan et al present a revised version of their manuscript on IO observations at two Indian stations located in East Antarctica over three summers. The manuscript has improved with additional literature and clarifications of the data. I have remaining questions regarding my previous review and additional suggestions.*

RESPONSE: We thank the reviewer for the comments and have made changes according to the specific comments below.

*If possible, it is encouraged to include a campaign average and stdev of the valid scans in the supporting information. It would be helpful to include this to understand the general range of the profiles during the campaign. In Ln 309-311, you mention that maximum IO mixing ratios were observed near 400m with a decrease above this. Does 400m correspond to a typical boundary layer height in this region? If so, please include this in the discussion.*
RESPONSE: We have now included a table in the supplementary text with the average and standard deviation for each campaign as requested (Table S1). We have also included a sentence regarding the boundary layer height as suggested (Line 311).

*In Ln 315-317, you mention that most models predict a strong IO gradient as the main source is from the surface within the model. Also, in Ln 422-423, a conclusion of "The vertical profiles confirmed past hypothesis of a source from the ground considering a sharp gradient" is made. However, Figure 8 seems like the IO is actually decreasing below ~ 400m towards the surface. Was this trend also observed in other valid vertical profile measurements of IO? It would be interesting to include a discussion of the possible explanations of the decrease in observed IO levels below the boundary layer towards the surface. As it is difficult to see the vertical profiles of IO close to the surface, it is suggested to include a graph similar to Figure 8 but with altitude < 1km in the supporting information.*
RESPONSE: While the 'high-resolution' plots show a decrease, there is a lack of enough information available from the radiative transfer model to conclude this. Indeed, considering the uncertainty in Figure 8, we cannot say that the decrease is significant. We would need more information for discerning the vertical profiles close to the surface and hence we do not offer a conclusion either way since the current method does not enable it. The number of datapoints along the vertical dimension in the Figure 8 are subsampled from a profile created by MAPA. Further information regarding the radiative transfer, and the information that can be got from it are given in the MAPA description paper.

*The references for each of the locations are still not added. Please add these in the figure caption.*
RESPONSE: Added.

*Based on the back trajectories in Figure 2, I'm surprised that there's very few airmasses (red dots in Fig. S10) that are categorized 'ocean airmass' as described in Ln 407-413. For example, during ISEA-35 at Bharati, airmasses are coming from all directions including the ocean. Could you add more details on how the categorization was made based on the back trajectories? In Fig. S10, as it is difficult to see which observation profile corresponds to each type of airmass, it is suggested to just add the airmass category (red dots) for the valid profiles.*
RESPONSE: The back trajectories care classified according to the heigh, in addition to the location. Although most air masses are 'oceanic', they pass over the ocean above 500 m, and hence are not considered as affected by oceanic emissions.

*Also, please Include time zone in the x axis (or top of each graph) for all the figures including figures 3, 4,5,6,7,8, S2, S3, S4, S5, S6, S7.*
RESPONSE: This is now made clear in all the figure captions.

**Additional Specific Comments:**

*(Ln 70) Please specify what the measured iodocarbons weren't able to explain high levels of. Was it IO/I2?*
RESPONSE: …of IO, added.

*(Ln 94) Please specify what measurement technique was used for the vertical profile of IO.*
RESPONSE: It was the MAX-DOAS instrument, this is now added.

*(Ln 96) Please specify what the detection limit and uncertainty of IO was for the cited study.*
RESPONSE: Added.

*(Ln 164) suggested to use 'large' SZA rather than 'high' SZA throughout the manuscript.*
RESPONSE: Changed.

*(Ln 183) change to "…the estimated value ranged between..."*
RESPONSE: Changed.

*(Ln 198) "using the" is used twice*
RESPONSE: Corrected.

*(Ln 279) Just to clarify, the range is for <30m? Please include the average and standard deviation.*
RESPONSE: The details are now added in Table S1.

*(Ln 348) " …shown to peak…"*
RESPONSE: Corrected.